# Formation of Citrazinic Acid Ions and Their Contribution to Optical and Magnetic Features of Carbon Nanodots: A Combined Experimental and Computational Approach

**DOI:** 10.3390/ma14040770

**Published:** 2021-02-06

**Authors:** Francesca Mocci, Chiara Olla, Antonio Cappai, Riccardo Corpino, Pier Carlo Ricci, Daniele Chiriu, Marcello Salis, Carlo Maria Carbonaro

**Affiliations:** 1Department of Chemistry and Geological Science, University of Cagliari, Cittadella Universitaria, I-09042 Monserrato, Italy; fmocci@unica.it; 2Department of Physics, University of Cagliari, Cittadella Universitaria, I-09042 Monserrato, Italy; chiara.olla@dsf.unica.it (C.O.); antonio.cappai@dsf.unica.it (A.C.); riccardo.corpino@dsf.unica.it (R.C.); carlo.ricci@dsf.unica.it (P.C.R.); daniele.chiriu@dsf.unica.it (D.C.); masalis@unica.it (M.S.)

**Keywords:** citrazinic acid, Raman spectroscopy, NMR spectroscopy, carbon nanodots, DFT calculations

## Abstract

The molecular model is one of the most appealing to explain the peculiar optical properties of Carbon nanodots (CNDs) and was proven to be successful for the bottom up synthesis, where a few molecules were recognized. Among the others, citrazinic acid is relevant for the synthesis of citric acid-based CNDs. Here we report a combined experimental and computational approach to discuss the formation of different protonated and deprotonated species of citrazinic acid and their contribution to vibrational and magnetic spectra. By computing the free energy formation in water solution, we selected the most favoured species and we retrieved their presence in the experimental surface enhanced Raman spectra. As well, the chemical shifts are discussed in terms of tautomers and rotamers of most favoured species. The expected formation of protonated and de-protonated citrazinic acid ions under extreme pH conditions was proven by evaluating specific interactions with H_2_SO_4_ and NaOH molecules. The reported results confirm that the presence of citrazinic acid and its ionic forms should be considered in the interpretation of the spectroscopic features of CNDs.

## 1. Introduction

Carbon nanodots (CNDs) are carbon-based nanoparticles which represent a new emerging field of research for optoelectronic and photonics purposes due to their unique optical properties. Among them stands out a stable and tunable photoluminescence in the visible region combined with cost-effective synthesis routes [1,2,3,4]. The origin of their fluorescent properties is still a debated topic, mainly because of the peculiar excitation dependence that in general causes a red shift of the emission as the excitation wavelength increases. The luminescence is currently interpreted for the most within the core-shell picture, assuming a crystalline carbon core and a disordered surface that could host different functional groups and molecular moieties [2,5,6,7]. These emitting core states, surface states and molecular states are expected to contribute to the observed luminescence and its excitation dependence. Clearly, the precise determination of each contribution is challenging, in agreement with the variable order-disorder ratio reported depending on the synthesis conditions, such as, for example, precursors and temperature [8,9].

A possible strategy requires the study of the complex chemical reactions of the involved precursors and the comparison of CNDs properties to the ones of the precursors or the products of reaction. This approach focuses on CND molecular state which was largely confirmed in previous works [10,11,12,13]. The molecule-like model was also supported by quantum-chemistry calculations, where different N-doped aromatic moieties were considered, together with their aggregated forms, as the origin of the optical properties of bottom-up synthesized CNDs [14,15,16,17,18].

Nowadays, CNDs are usually obtained from thermal degradation of organic molecules among which the most common precursor is citric acid. The thermal degradation of citric acid produces few fluorescent intermediates which can be responsible for the luminescence properties of citrate-based CNDs [19], promoting a detailed investigation of these intermediate species to deepen the comprehension of CND optical and structural properties. In particular, recent studies on highly efficient blue-emitting CNDs obtained from the thermal decomposition of citric acid and nitrogen sources have recently ascertained the presence of fluorescent citrazinic acid (CZA, 2,6-dihydroxyisonicotinic acid) and its derivatives [8,20,21,22,23,24].

The specific attribution of the observed emission properties to CZA or to other fluorophores is still a discussed topic. Strauss et al., claimed the formation of two types of molecular fluorophores during the synthesis of their citrate-urea CNDs but no evidence of CZA resulted from their NMR measurements [25]. On the other hand, Sharma et al., reported the opposite result, emphasizing the resemblance between the NMR spectra of their CNDs with that of CZA [5]. A similar evidence corroborated the assignment of the gathered properties to a 2-pyridone derivative [26].

CZA is an organic fluorophore, consisting of a 2,6-dihydroxypyridine ring with a carboxylic group in position 4, which applications ranges from the production of photosensitizers to the synthesis of new microporous materials [27,28,29]. Its optical characteristics depend on both aggregation states and chemical environment: CZA presents the tendency to form aggregation states as dimers at high concentrations and tautomeric forms according to the solution pH [30,31]. Despite its potential role in the CND emission properties, only few research groups investigated its physico-chemical features in detail.

Sarkar et al., compared the simulated Raman spectra of the possible tautomeric species to the experimental results obtained in acidic and basic solutions stating the coexistence of different molecular forms varying the pH environment [32]. As the CZA concentration increases, weakly fluorescent J-type aggregates can be formed from the keto tautomer of CZA [30] which undergoes protonation or de-protonation under extreme pH conditions [31], further expanding the application of citric acid related CNDs to pH sensing [24].

In this work we propose a synergic theoretical and experimental approach, starting from a systematic computational study of all the tautomeric forms present in aqueous solutions to calculate their formation energy. Then, the simulated vibrational and magnetic features of formed species are compared with the experimental Raman and NMR spectra of CZA aiming to identify the possible contribution of the latter to the properties of N-doped citric acid related CNDs.

## 2. Materials and Methods

Citrazinic acid (purity 0.97, Sigma-Aldrich, St. Louis, MO, USA) and milli-Q water were used as received without further purification. Raman spectra were recorded on a solution of citrazinic acid (CZA) in water with concentration of 10–50 mg/L.

Surface Enhanced Raman spectroscopy (SERS) measurements were obtained in back scattering geometry with a Micro Raman scattering confocal system (SOL Confotec MR750, SOL instruments Ltd., Minsk, Republic of Belarus) equipped with Nikon Eclipse Ni microscope (Nikon Instruments Europe BV, Amsterdam, The Netherlands). The samples were excited at 532 nm (IO Match-Box series laser diode) and collected with a spectral resolution of 0.6 cm^−1^ (acquisition time 20 s, number of acquisition 5, sensor temperature −24 °C, objective Olympus 50x, grating with 1200 grooves/mm, power excitation 3 mW) SERS supports were ITO glasses coated with silver nanoparticles (S-Silver SERS substrates, Sersitive, Warsaw Poland). 

All the quantum-chemistry calculations were performed by using the Gaussian 16 suite of programs [33]. We assumed as starting geometry the keto tautomer of citrazinic acid which, from previous calculations [30], resulted to be more stable in water solution than the imine tautomer.

Geometry optimization down to the Self Consistent Field (SCF) energy of each investigated species of CZA was performed by means of DFT calculations carried on at a B3LYP/6311 + + G(d,p) [34,35] level of theory. The simulations were carried out in vacuum and with different solvents, such as water, ethanol (EtOH), N,N-Dimethylformamide (DMF) and Dimethylsulfoxide (DMSO). The interaction of CZA molecules with solvents was accounted for by applying the Self Consistent Reaction Field (SCRF) model and simulating the dielectric solvent through the Polarizable Continuum Model (PCM) calculation within the integral equation formalism (IEFPCM) [36]. All the optimized structures were verified to be real energy minima with no imaginary frequencies in the vibrational spectra.

UV absorption spectra were simulated in terms of vertical energy transitions with the solvent environment assumed to be the same of the ground state of the molecule by using both the functional with the same basis set. NMR features were computed from non-dynamic single point SCF minimum structure with Gauge-Independent Atomic Orbital (GIAO) method [37].

## 3. Results

In the following sections we report the free energy formation of different ions of CZA calculated according to the proper thermodynamic cycles. The gathered minimum energy tautomers and rotamers are then exploited to interpret the experimental Raman and NMR spectra. Optical properties in terms of pH conditions are also modelled by calculating the potential energy surface of the interacting molecular system.

### 3.1. Thermodynamic Cycles

We carried out DFT calculations on either the neutral or protonated/de-protonated forms of CZA, starting from the keto tautomer already reported as the most favorite one with respect to the imine form and considering all the possible protonation and de-protonation processes (i.e., accounting for all the possible tautomers and rotamers when adding or removing a proton). Ball and sticks sketches of each species in water solvent are listed in Table 1. In the table we reported for each species the tautomer and the rotamer with the lowest SCF solvation free energy. In the case of the single de-protonated CZA molecule we reported the two tautomers with the lowest solvation free energy (CZA1M and CZA1MA), since the release of a proton in solution from the carboxylic or pyridinic site is relevant for the experimentally reported pK_a_ values. 

By following the computational scheme proposed by Liptak et al., [38] we also evaluated the free energy of acid-base reaction in water solution (*ΔG_sol_*) of each species with respect to the neutral CZA assuming an acid-base equilibrium in water and performing the required calculations according to the thermodynamic cycle 1 and 2 [39] reported in Figure 1. Analogous cycles were considered for the double and triple deprotonated species CZA2M, CZA3M, and for the double protonated species CZA2P (see Appendix A for a detailed description of the calculation). 

To reach the required level of accuracy, the free energies of hydration of H^+^, H_2_O, OH^−^ and H_3_O^+^ were taken from the experiments [40]. Considering the proper cycle for each species, we predicted the energy required to CZA to release or acquire one or more protons in/from water. As reported in Table 1, all the de-protonated and protonated CZA species are less energetically stable with respect to the neutral form. We did not report the evaluation of the de-protonation at the NH site because it was more than 10 times less favorable than on the OH and COOH ones. Table 1 also reports the evaluated pK_a_ for each reaction, showing a good agreement with the experimental values of the carboxylic and the phenolic group of CZA. The experimental pH value of a 10 mg/L of CZA was measured to be 4.0, leading to a pK_a_ of 3.81, in good agreement with the value calculated for the CZA1MA species. To discuss SERS and NMR features we considered the species with the lowest solvation free energies.

### 3.2. Potential Energy Surface (PES) Simulations

The results of the calculated thermodynamic cycles clearly indicate that the protonated and de-protonated species can be formed under proper pH conditions. To simulate the formation of these charged molecules we considered the interaction of CZA with H_2_SO_4_ and NaOH in water. In both cases we performed relaxed scan calculations to probe the potential energy surface (PES) for the interaction along a specific interaction coordinate. We selected as possible trajectories the ones leading to the interaction of the H_2_SO_4_ and NaOH molecules with the phenolic group. In the case of NaOH, the selected trajectory considered the distance between the H atom of NaOH and the H atom of the hydroxyl group in the CZA molecule. For the H_2_SO_4_ case we selected the distance between a H atom of H_2_SO_4_ and the O atom of the phenyl ring of CZA. As reported in Figure 2, the two calculated PES show a minimum of energy achieved when the CZA molecule and the probing molecule exchange a hydrogen atom, leading to the formation of the protonated and de-protonated species in the case of interaction with H_2_SO_4_ and NaOH respectively. As depicted by the molecular sketches reported in the figure, representing the molecules at the starting interaction distance and at the minimum interaction distance, the hydrogen of the H_2_SO_4_ is acquired by the CZA molecule leaving a H_2_SO_4_ anion in solution, whilst in the case of interaction with NaOH, the CZA molecule loses one proton and the reaction produces water and a Na cation. Although with rough approximations, the reported trajectories represent what is expected under proper pH conditions. We also verified that the optical absorption calculated in the minima reproduces the optical absorption of the protonated and de-protonated species, as expected.

### 3.3. Optical Properties

We calculated the optical absorption features of the different species, and for the neutral CZA we tested the effect of different solvents. The results are reported in Table 2 for the computed Highest Occupied Molecular Orbital (HOMO) to the Lowest Unoccupied Molecular Orbital (LUMO) transitions.

The absorption of neutral CZA is calculated at 347.4 nm and it is both blue and red shifted depending on the charge of the considered CZA ion. The calculated oscillator strengths, however, do not change very much from one species to the other. As for the solvatochromism, the recorded variations of the HOMO-LUMO absorption are quite small, within a few nanometers, showing an hypsochromic shift for polar protic solvents (water and EtOH) with respect to the bathochromic shift for polar aprotic solvents (DMSO and DMF). 

### 3.4. Vibrational Analysis

The experimental spectra were recorded by exploiting the interaction of the molecules with a nanostructured silver surface to produce the SERS signal. Under these circumstances an enhancement up to 10^6^ was previously reported for CZA [32], suggesting that the contribution of even the less favoured CZA ions, down to the CZA2M, could be detected. We compared in Figure 3 and Figure 4 the experimental SERS spectrum with the Raman vibrations calculated for each considered species. The calculated modes were scaled according to a quadratic scaling factor, as proposed by [41]. As expected, the vibrations of the four considered species are quite similar, their contemporary presence eventually contributing to the experimental spectrum according to their relative amount in solution. The main peaks are listed in Table 3 with the assignment of the corresponding computed vibration modes as deduced from the neutral CZA molecule. The table also illustrates by blue arrows the motion direction of various atoms for each mode.

The experimental 438 cm^−1^ peak can be assigned to an in-plane rocking vibration, as well as the 1269 and 1337 cm^−1^ peaks. The 1269 cm^−1^ peak is also partially due to in-plane scissoring oscillations. The vibration at 522 cm^−1^ is related to the combination of out of plane twisting motions and to symmetric stretching of the phenyl ring. Out of plane wagging oscillations and in plane scissoring motions produces the peak at 602 cm^−1^, as well as the band at 678 cm^−1^ (scissoring) and at 775 cm^−1^ (wagging). Finally, the large band peaked at 1517 cm^−1^ can be related to asymmetric stretching vibrations within the phenyl ring.

Concerning the contribution of the other species it can be inferred from some sharp features overlapped to the main bands whose positions may be the fingerprint of the presence of a specific species. To find these signatures we considered the small number of peaks that could not be assigned to the contribution of the neutral form. This is the case of the lines at 706, 1117 and 1750 cm^−1^ which could be ascribed to the presence of the CZA1P species, whilst the CZA1MA is mainly individuated in the signals at 985, 1267 and 1513 cm^−1^, the first one in sharing with the CZA2M and the other ones with the CZA1M. Finally, the 1391 cm^−1^ line can be assigned to both the CZA1M and CZA1P species.

### 3.5. NMR Calculations

Accounting for the experimental sensitivity of NMR technique, we calculated the NMR features of CZA, CZA1M and CZA1MA species only. The variations of the isotropic shielding constant calculated with different PCM solvent models (like water and DMSO) are negligible, with a variation within 0.1 ppm for ^13^C. To estimate the NMR signals, based on the previous considerations on the formation energies of different species and their rotamers and tautomers, we considered the two rotamers of neutral CZA and the two tautomers of the single de-protonated molecules, CZA1M and CZA1MA. The computed isotropic shielding constants were re-scaled to chemical shift by means of the experimental spectrum of CZA in DMSO [25], using as reference value the chemical shift of the carboxylic carbon. We assumed fast exchange in the ^13^C chemical shift scale among the rotamers of each tautomer, thus providing 4 NMR signals referred to different carbons as reported in Table 4. This assumption is consistent with the observation of four main peaks in the experimental spectrum of CZA, as also confirmed by data provided by the producer of CZA powders. The C ring atoms in positions 2 and 6 were indicated as C-carbonyl/hydroxyl to account for the two different chemical environments in fast exchange, the same also holds for the C atoms in positions 3 and 5, referred as C-β-carbonyl/hydroxyl in the table. As reported, there is in general a good agreement between experimental and calculated chemical shift (see discussion for details). Similar results were also obtained by scaling the isotropic shielding to the values of a computed reference, namely tetra-methyl-silane (TMS). Those values are reported within parenthesis in the table and agree quite well with the previous scaling. In addition, we compared the computed ^1^H chemical shift with the experimental one recently reported by our group [31], re-scaling the calculated values according to the ones obtained for the ^1^H in TMS. 

The ^1^H experimental spectrum reports two signals. The first one is a broad peak at about 12.1 ppm related to the contribution of 3 H atoms bonded to heteroatoms (carboxylic proton, hydroxyl group and pyridinic proton). The second sharp peak at 6.2 ppm is assigned to two aromatic H atoms bonded to C atoms in ortho positions. The values for the three considered species are also reported in Table 4. The maximum split among the signal is obtained for the CZA1MA case (1.2 ppm). Considering the large CZA concentration in the NMR experiment (4 mg/mL) we also calculated the NMR signals in the case of neutral CZA dimers, accounting for the most energetically favourable ones [42], namely linear head-to-tail, tail-to-tail and head-to-head dimers, in decreasing order of formation energy (head of the molecule is the N atom, the tail the carboxylic group). Schematic representation and calculated NMR signals (rescaled to TMS ones) are reported in Table 5. The formation of the aggregates causes a de-shielding of the protons bonded to the heteroatoms leading to a maximum split of 2.5 ppm for the head-to-tail dimer (the most energetically favoured).

The effect on the ^1^H signal due to the hydrogen bonding to solvent molecules cannot be properly taken into account with a PCM solvent. Therefore, to verify such effect on the dominating species of CZA in solution, the interactions with the DMSO solvent were modelled adding a solvent molecule to the system and calculating the chemical shift on the structure of such systems embedded in the PCM solvent. The chemical shift of the H atoms involved in the hydrogen bond are reported in Table 6. The chemical shift of the H atoms not involved in the H-bonding is only slightly affected by the interactions (<0.3 ppm). Details on the NMR calculations are reported in the Appendix A.

## 4. Discussion

The calculated formation energies of the different CZA ions in water disagree with a previous report [32], where the most favoured species was predicted to be the protonated one. In addition, the formation of other species considered in that paper costed more than 10 eV, thus virtually excluding their formation unless extreme pH conditions are considered. On the contrary, according to our results, under physiological pH conditions (in the range 3–9, being the standard pH of CZA acid), the neutral species is the most abundant one, and some content of CZA ions can be expected, being the difference in energies with the neutral species below 4 eV (93 kcal/mol, see Table 1). It is also clear that extreme pH conditions, achieved by the addition, for example, of H_2_SO_4_ acid or NaOH base [31], are required to favour the formation of the fully protonated and fully de-protonated species. This is also confirmed by our PES calculations, that despite the approximation of selected specific interaction trajectories, reproduce what is expected under acidic and basic pH conditions. We also verified that the optical absorptions calculated in the minima (here not reported for the sake of brevity) are consistent with the optical absorptions of the protonated and de-protonated specie, as expected [31,32].

We calculated the HOMO-LUMO transitions of the investigated CZA species to evaluate if the observed absorbance characteristics of CNDs, such as the blue shift as the concentration increases, the pH dependence and the solvatochromic effect reported in the literature, could be related to the optical properties of CZA. As we can observe from Table 2 the absorption transition typically observed at about 340–350 nm in CNDs can be related to the presence of neutral CZA molecules, as already reported [31]. The presence of different CZA ions could also shift the absorption peak. However, accounting for the relative population of CZA ions under physiological pH conditions, only the neutral and the singly de-protonated ones can contribute significantly to the observed optical properties (see Appendix A). It is interesting to note that the CZA1MA species could be responsible of the shoulder at about 400 nm typically observed in CNDs. However, the experimentally recorded spectra at low CZA concentration are peaked at about 350 nm, as the one of CNDs, thus suggesting that the experimental pH conditions promote the CZA+CZA1M equilibrium. To single out the optical features of protonated and de-protonated species one needs to shift the equilibrium to large acidic or basic pH, confirming previous results [31]. We also point out that the formation of the two single de-protonated CZA species, namely CZA1M and CZA1MA, can reproduce the experimental absorbance spectrum recorded under basic conditions and previously assigned to the fully de-protonated molecule (CZA3M) [31]. These findings also support the hypothesis that the recorded changes as a function of the CZA concentration under physiological pH conditions should be related to the formation of aggregates instead of to the presence of different CZA ions in solution [14,20]. We may speculate, however, that the charged species can work as reactive seeds for the nucleation of emitting centers, thus further broadening the near UV absorption band of CNDs. Concerning the solvatochromism, despite the small shift calculated, the reported data agree with the experimental spectra [5], confirming that the reported experimental shift in CNDs could be related to the properties of CZA.

The presence of the ionic species of CZA is confirmed by the SERS spectrum under physiological pH conditions. Indeed, the experimental spectrum is quite reach in the explored 350–1800 cm^−1^ range and can be easily interpreted assuming the presence of the four ionic species of CZA in water solution. The enhancement granted by the SERS technique allowed us to individuate the contribution of each species, thus confirming the presence of de-protonated and protonated CZA ions. We could also assign the main peaks to the vibration mode of the CZA molecule. These vibrations are clearly in agreement with the experimental ones of CNDs, because of the presence of aromatic rings, N-doping and functional groups, such as hydroxyl and carboxyl groups. In this perspective, vibrational spectroscopy can support but not exclusively confirm the molecular model of CNDs. We want to underline that there is some difference in the attribution previously proposed for the ionic species [32] which could be ascribed, in our opinion, to the different scaling factor adopted, a quadratic scaling function in the present case for all the species, a species-dependent factor in the other case, and to the theoretical level applied. 

To discuss if molecular features in CNDs can be deduced from NMR spectra, we calculated the NMR signals of CZA species in DMSO solvent and compared them to experimental results. Concerning the data of ^13^C, our findings agree within 2–3 ppm with the reference experimental data except for the signals of the C-β-carbonyl/hydroxyl of CZA1MA (about 10 ppm of difference) and the C-γ of the CZA1M and CZA1MA species (about 10 and 8 ppm respectively). No matches were found in the computed features for the two experimental chemical shifts at 125.9 and 34.3 ppm. The data re-scaled to the computational TMS reference also agree quite well with the experimental ones, with a mean larger discrepancy of about 10–15 ppm but confirming the general picture. 

As for the ^1^H-NMR spectrum of CZA, it is in agreement with the one reported by Strauss and co-workers [25], except for a shift of the large peak of the hetero-atoms proton of about 3 ppm. The sharp peak at 6.2 ppm is well reproduced by the calculated signals, also showing a shielding when de-protonated species are formed, in agreement with the experimental findings. On the contrary, the large peak is poorly reproduced when modelling the solvent with the PCM model alone, being at maximum calculated at 7.2 ppm. To reproduce the shielding is necessary to consider the solvent explicitly, and to consider that experimental data are collected with concentrated samples: in our experiment the CZA concentration was 4 mg in 1 mL of DMSO, about 3 order of magnitude larger than the concentration considered for absorbance features. Under these conditions the formation of CZA aggregates should also be considered. We indeed calculated the NMR signals of three selected dimers, the ones with lower formation energies [42]. The analysis of the simulations clearly indicates that the formation of aggregates produces a de-shielding effect on the signal assigned to protons bonded to hetero-atoms thus causing a low-field shift up to 9 ppm, in good agreement with the experimental data [25,31]. On the other hand, by considering the effect of the solvent explicitly, a larger de-shielding is observed, up to 14 ppm, so that the two effects, the formation of aggregates and interaction with the solvent, could contribute to the recorded peak at about 9–12 ppm. Two more considerations can be drawn from the detailed analysis of the chemical shift (see Appendix A). The first one is that the protons contributing most to the shift are the ones involved in the interaction among the two CZA monomers, thus supporting the idea that the formation of larger aggregates, as can be expected because of the large concentration, could produce a larger shift. The second one is that the largest de-shielding is calculated for the carboxylic hydrogen of the head-to-tail dimer (the most favoured one) suggesting that NMR measurements in non-protic solvents could detect the formation of CZA aggregates. Comparing the present results with the ones of literature we see that the reported magnetic signals of CZA can explain the ones reported for citric acid related CNDs, depending on the synthesis conditions. It is known, indeed, that, with reference to hydrothermal synthesis, a long reaction time favours the formation of aromatic domains [8] whilst low temperature promotes the formation of molecules [9,21]. Thus, the reported data could explain the features recorded for synthesis at low temperature (below 200 °C) and reduced reaction time [5,26]. We point out that, as reported by [25], low temperature syntheses are required to record the formation of molecular species and the CZA intermediate in the case of citric acid and urea synthesis. Besides, the synthesis method and the reaction time should also be considered, the presence of solvent or reactive atmosphere being important parameters. All these factors interactively contribute to the formation or not of specific molecular species. In general, longer time and higher temperature favour the formation of aromatic domains versus molecular ones [8]. Indeed, in the case of [25] a reaction at temperature larger than 150 °C for 20 min in microwave produced CNDs with no sign of CZA. On the contrary, synthesis at 200 °C for 10 min or at 180 °C for 1 h with heating mantle or reflux respectively allowed the observation of CZA or 2-pyridone related NMR signals [5,26]. A complete study of reaction temperature and time with different synthesis methods with respect to the formation of CZA is an interesting perspective that would be considered for a further research.

## 5. Conclusions

CZA is a fluorescent molecule that could explain some spectroscopic properties of CNDs produced by bottom-up synthesis of citric acid and N containing precursors, such as urea or other amides. To discuss if such a molecule is embedded within the structure of CNDs, we considered the reactions of CZA in water to assess the formation of specific ionic form of CZA and their solvation free energies, here calculated for the first time by quantum chemistry simulations. The presence of the protonated and de-protonated species was confirmed by Raman spectroscopy and NMR spectra. In the latter case we also considered the formation of aggregates to reproduce the ^1^H NMR signals. Both the vibrational and magnetic features here reported are observed in the case of CNDs, depending on the synthesis conditions (mainly kind of precursors, their relative concentration and temperature) thus supporting the idea that CZA molecule can participate to the optical and magnetic properties of citric acid related CNDs synthesized at low temperature and with short reaction time.

## Figures and Tables

**Figure 1 materials-14-00770-f001:**
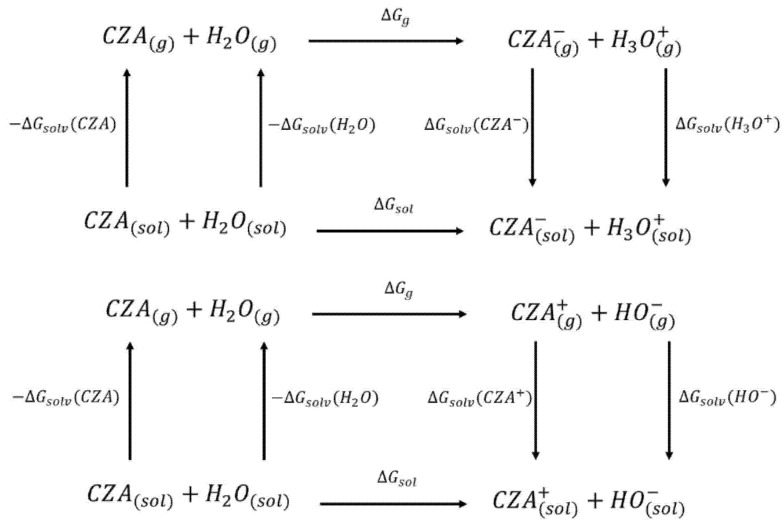
Thermodynamic Cycle 1 (**top**) and cycle 2 (**down**) used for the pK_a_ estimation. CZA^−^ represent the de-protonated form of CZA (either CZA1M or CZA1MA in Table 1) and CZA^+^ is the cation obtained by protonation of CZA (CZA1P in Table 1).

**Figure 2 materials-14-00770-f002:**
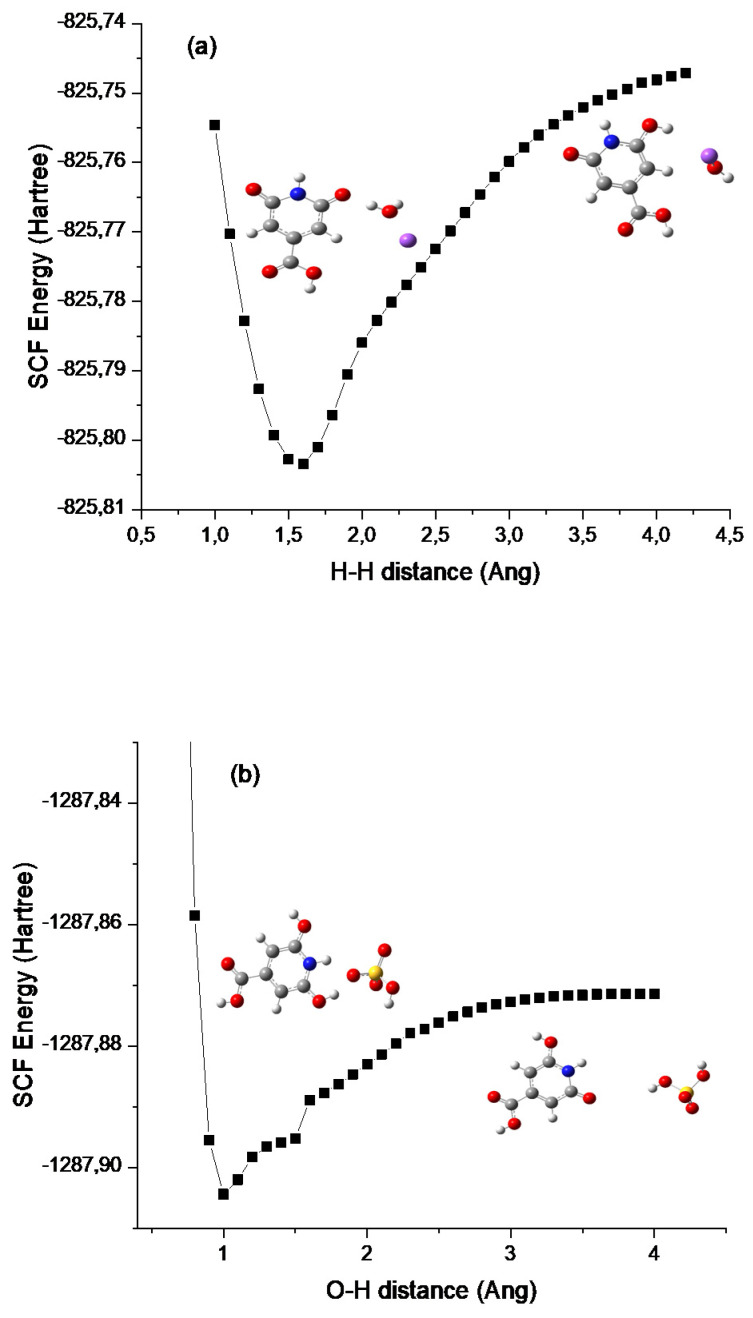
PES of CZA and NaOH (**a**) and H_2_SO_4_ (**b**) interactions. The insets report ball and stick schematic representations of the molecules at the starting position of the selected interaction trajectory and at the minimum of SCF energy (H = white atom, O = red atom, C = grey atom, N = blue atom, S = yellow atom).

**Figure 3 materials-14-00770-f003:**
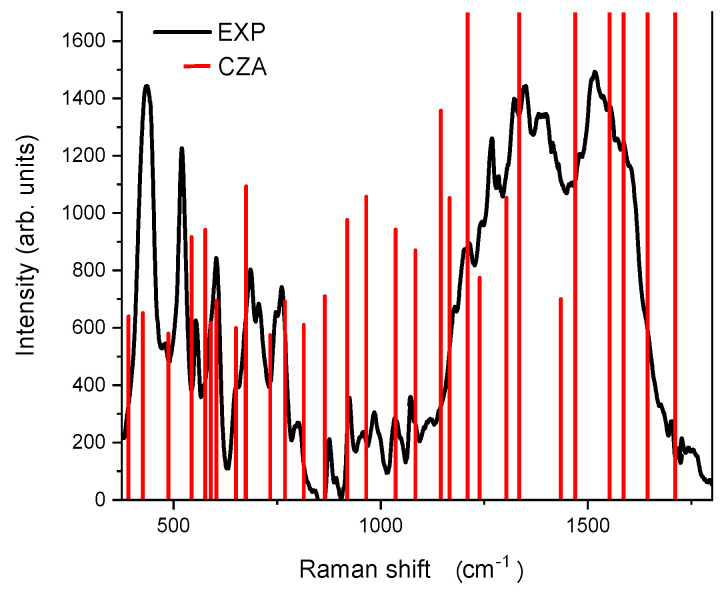
Experimental SERS and computed Raman vibrations for CZA case.

**Figure 4 materials-14-00770-f004:**
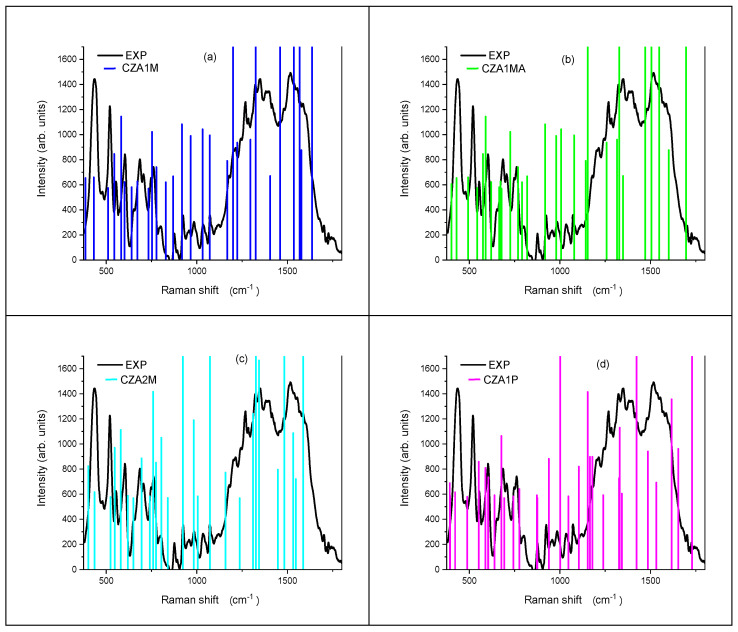
Experimental SERS and computed Raman vibrations for (**a**) CZA1M; (**b**) CZA1MA; (**c**) CZA2M; (**d**) CZA1P.

**Table 1 materials-14-00770-t001:** List of the minimum energy tautomer of CZA and of its protonated/de-protonated forms (q is the charge state) with computed *ΔG_sol_* (see Figure 1) and pK_a_ values in water. A ball-and-stick representation is reported for each species (H = white atom, O = red atom, C = grey atom, N = blue atom).

Sample ID	q	Structure	ΔG_sol_ (kcal/mol)	pK_a_
CZA	0	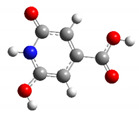	0	-
CZA1M	−1	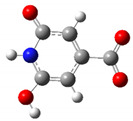	18.2	7.1
CZA1MA	−1	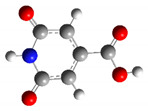	13.4	3.5
CZA2M	−2	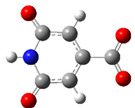	37.2	21.0
CZA3M	−3	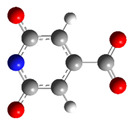	91.4	60.7
CZA1P	+1	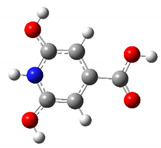	31.5	16.8
CZA2P	+2	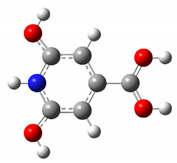	87.7	88.0

**Table 2 materials-14-00770-t002:** List of HOMO-LUMO transition and oscillator strength for different species in water, and for CZA in different solvent.

Species	HOMO-LUMO (nm)	Oscillator Strength
CZA CZA1M CZA1MA CZA2M CZA1P	347.4 307.4 410.5 340.1 305.5	0.1197 0.1374 0.1201 0.1738 0.1566
CZA—DMSO CZA—DMF CZA—EtOH	351.0 351.1 350.4	0.1296 0.1303 0.1260

**Table 3 materials-14-00770-t003:** Experimental and calculated vibrations of CZA molecule, and pictorial representations of the normal modes (H = white atom, O = red atom, C = grey atom, N = blue atom; arrows indicate the relative intensity and the direction of motion of each atom).

Experimental (cm^−1^)	Calculated (cm^−1^)	
438	425	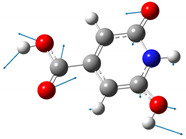
522	487	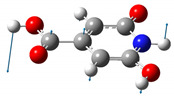
522	543	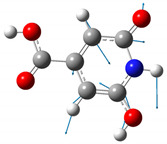
602	590	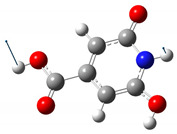
602	603	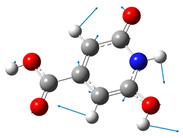

**Table 4 materials-14-00770-t004:** List of calculated and experimental NMR chemical shifts in ppm of ^13^C and ^1^H atoms in CZA species.

C-Type	CZA	CZA1M	CZA1MA	Exp
C-carbonyl/hydroxyl C-β-carbonyl/hydroxyl C-γ C-carboxyl	159.4 (166.5) 98.8 (105.9) 142.7 (149.7) 166.6 (173.6)	158.2 (166.2) 96.3 (104.2) 155.5 (163.4) 166.6 (174.5)	162.4 (173.2) 88.7 (99.4) 137.9 (146.7) 166.6 (177.4)	162.0 [25] 98.6 [25] 145.1 [25] 166.6 [25]
**H-Type**	**CZA**	**CZA1M**	**CZA1MA**	**Exp**
Aromatic-H Hetero-H	6.5 7.2	6.5 6.5	5.7 6.9	6.2 [31] 12.1 [31]

**Table 5 materials-14-00770-t005:** List of calculated NMR chemical shifts for ^1^H atoms in CZA dimers (H = white atom, O = red atom, C = grey atom, N = blue atom).

Dimer Type	Aromatic-H	Hetero-H	
Head-to-Head	6.5 ppm	8.1 ppm	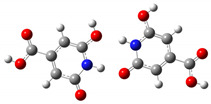
Tail-to-Tail	6.5 ppm	8.9 ppm	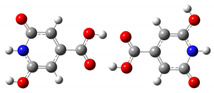
Head-to-Tail	6.6 ppm	9.1 ppm	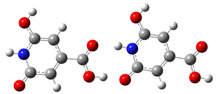

**Table 6 materials-14-00770-t006:** Calculated NMR chemical shifts for ^1^H atoms involved in the hydrogen bond with a DMSO molecule (H = white atom, O = red atom, C = grey atom, N = blue atom, S = yellow atom).

H-Bonding Group	Chem. Shift	
COOH	14.3 ppm	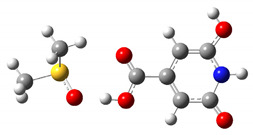
NH	13.8 ppm	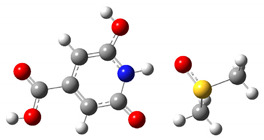
OH	13.5 ppm	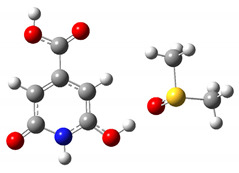

## Data Availability

Data available on request due to privacy restrictions.

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
