# Peer review of "Formation of Citrazinic Acid Ions and Their Contribution to Optical and Magnetic Features of Carbon Nanodots: A Combined Experimental and Computational Approach"

_materials, 2021, doi:10.3390/ma14040770_

Round 1

Reviewer 1 Report

In this work by Mocci et al., the authors investigated how citrazinic acid ions can be (de)protonated, which, in turn, affects the optical properties of the material. It is definitely an interesting topic, which fits well the scope of Materials. In general, I have no major objections regarding the described findings. However, some errors and claims should be corrected before the paper can be recommended for publication.
1) "Citrazinic acid (purity 0.97, Sigma-Aldrich)" (Line 83) - what is purity 0.97?
2) More parameters regarding Raman characterization e.g. integration number and time, as well as the type of the objective should be reported.
3) Structures do not fit in the rows of Table 1, which interferes with the border formatting. Please correct it in this table and Tables 3, 5, and 6.
4) Typo at Line 197 - the decimal point should be ".".
5) The discoveries done in this research may be helpful for the explanation of the optical properties of carbon nanodots (CNDs) obtained from such precursors. However, my main concern is that these claims are not validated in this article. After reading the abstract and the title, a reader may have a feeling that the article contains a synthetic part, in which these suspicions would be verified.

The authors stress many times that such a chemical compound may explain some spectroscopic properties of CNDs produced from such a feedstock, but there is not a single synthesis of CNDs reported in this work, which would support it.

I invite the authors to provide a convincing explanation regarding this issue.

Reviewer 2 Report

Comments on Formation of citrazinic acid ions and….carbon nanodots

The authors report vibrational and magnetic resonance spectra of various forms of citrazinic acid (CZA) in water, and deploy computational modeling to help interpret the data. The computational model – the B3LYP density functional represented in a flexible Pople basis 6311++G(d,p) – is a reasonable choice. For this organic system. The solvent is represented mostly by a polarizable continuum model, though for some issues in the NMR an explicit solvent molecule is chosen to interact with the CZA species.

The relative energies of variously charged variants of CZA are estimated with the help of thermodynamic cycles; the most stable of these is the neutral form. Deprotonated forms in which H(+) is removed either from the carboxyl group or a ring hydroxyl group lie within 20 kcal/mol of the neutral. A cationic form is found in which the ring carbonyl is protonated is about 30 kcal less stable than the neutral CZA. I had imagined that deprotonation or protonation of the ring >NH might have been overlooked; but my calculations suggest both of these alternatives are well out of reach energetically.

Limited exploration of the PES for approach of H2SO4 and NaOH suggest that proton transfer is feasible when the pH is extreme; computed (TD-DFT, in PCM) spectra for the proton-transfer complexes are said to be consistent with those of the isolated charged CZA species.

TD-DFT spectra show considerable shifts from the 350 nm first transition of CZA upon (de)protonation, but solvent effects on the spectrum for neutral CZA are minor. .The computed IR spectrum for CZA accounts for much of the spectrum observed for pH near 7, but some features of the experimental spectrum can be attributed to charged species. Simulation of the proton NMR spectra is less successful for the aromatic XH region, but consideration of interaction of explicit solvent molecules and dimerization of CZAs suggests that deshielding can be attributed to aggregation.

The description of relative stability of species in this report is not consistent with the work cited as reference 32. Unfortunately the authors of ref. 32 do not include coordinates for molecular structures nor DFT total energies in their report or supplementary information. These I think would have settled the issue, along with a comparison of thermodynamic cycles employed in the two projects

This work is carefully designed and executed, and clearly reported. While I would like to see a more complete explanation for the discrepancy just mentioned above, this unresolved difficulty should not delay publication.

Minor details:

Caption, Fig 1: Can it be that CZA¾ would represent the deprotonated form(s) of CZA? The negative sign is indistinct in my pdf.

Line 244: CZA rather than CZ

Line 249: molecules rather than molecule

Line 308: perhaps “required” rather than “requested”

Line 357: perhaps “except for” rather than “but for” (the latter is certainly correct English usage, just less familiar) See also line 364.mulation of the

Round 2

Reviewer 1 Report

Thank you for your reply. However, the questions I asked are not to help myself but the readers. Consequently, the explanation to Remark #1 should be added to the text. Moreover, headlines should not be separated from corresponding sections (Line 165). These comments can be handled at the proof stage. I recommend the publication of the article.